# Integrating Clinical and Histopathological Data to Predict Delayed Graft Function in Kidney Transplant Recipients Using Machine Learning Techniques

**DOI:** 10.3390/jcm13247502

**Published:** 2024-12-10

**Authors:** Sittipath Tirasattayapitak, Cholatid Ratanatharathorn, Sansanee Thotsiri, Napun Sutharattanapong, Punlop Wiwattanathum, Nuttapon Arpornsujaritkun, Kun Sirisopana, Suchin Worawichawong, Lionel Rostaing, Surasak Kantachuvesiri

**Affiliations:** 1Division of Nephrology, Department of Medicine, Faculty of Medicine Ramathibodi Hospital, Mahidol University, Ratchathewi, Bangkok 10400, Thailand; sittipath01@outlook.co.th (S.T.); sansanee.tht@mahidol.ac.th (S.T.); napun.sut@mahidol.ac.th (N.S.); Punlop.wiw@mahidol.ac.th (P.W.); 2Excellence Center for Organ Transplantation, Faculty of Medicine Ramathibodi Hospital, Mahidol University, Ratchathewi, Bangkok 10400, Thailand; nuttapon.apo@mahidol.ac.th (N.A.); kun.sir@mahidol.ac.th (K.S.); suchin.wor@mahidol.ac.th (S.W.); 3Department of Clinical Epidemiology and Biostatistics, Faculty of Medicine Ramathibodi Hospital, Mahidol University, Ratchathewi, Bangkok 10400, Thailand; cholatid.rat@mahidol.edu; 4Vascular and Transplant Unit, Department of Surgery, Faculty of Medicine Ramathibodi Hospital, Mahidol University, Ratchathewi, Bangkok 10400, Thailand; 5Division of Urology, Department of Surgery, Faculty of Medicine Ramathibodi Hospital, Mahidol University, Ratchathewi, Bangkok 10400, Thailand; 6Department of Pathology, Faculty of Medicine Ramathibodi Hospital, Mahidol University, Ratchathewi, Bangkok 10400, Thailand; 7Nephrology, Hemodialysis, Apheresis and Kidney Transplantation Department, University Hospital Grenoble, 38000 Grenoble, France

**Keywords:** deceased-donor kidney transplantation, delayed graft function, machine learning prediction, time-zero kidney biopsy, kidney transplantation

## Abstract

**Background:** Given the significant impact of delayed graft function (DGF) on transplant outcomes, the aim of this study was to develop and validate machine learning (ML) models capable of predicting the risk of DGF in deceased-donor kidney transplantation (DDKT). **Methods:** This retrospective cohort study was conducted using clinical and histopathological data collected between 2018 and 2022 at Ramathibodi Hospital from DDKT donors, recipients, and post-implantation time-zero kidney biopsy samples to develop predictive models. The performance of three ML models (neural network, random forest, and extreme gradient boosting [XGBoost]) and traditional logistic regression on an independent test data set was evaluated using the area under the receiver operating characteristic curve (AUROC) and Brier score calibration. **Results:** Among 354 DDKT recipients, 64 (18.1%) experienced DGF. The key contributing factors included a donor body mass index > 23 kg/m^2^, donor diabetes mellitus, a prolonged cold ischemia time, a male recipient, and an interstitial fibrosis/tubular atrophy score of 2–3 in the time-zero kidney biopsy sample. The random forest model had a specificity of 99.96% and an AUROC of 0.9323, the neural network model had a specificity of 97.43% and an AUROC of 0.844, and the XGBoost model had a specificity of 99.81% and an AUROC of 0.989. A traditional statistical model had a specificity of 84.4% and an AUROC of 0.769. **Conclusions:** Predictive models, especially XGBoost models, have potential as tools for assessing DGF risk post-DDKT, guiding acceptance decisions, and avoiding risky biopsy, and they may be crucial in resource-limited settings.

## 1. Introduction

Kidney transplantation (KT) is an effective intervention for end-stage kidney disease (ESKD) that results in decreased mortality rates and enhanced quality of life. It is crucial that optimal kidney function is achieved immediately after KT, as characterized by the urine output exceeding 1 L/day within 24 h and a consistent reduction in serum creatinine levels (>1 mg/dL per day) that results in a serum creatinine level < 3 mg/dL by the end of the first week [1]. This is known as immediate graft function.

Delayed graft function (DGF) occurs when the transplanted kidney fails to immediately function at an adequate level, which necessitates the application of renal replacement therapy within the first week post-transplant [2,3]. According to the US government’s Scientific Registry of Transplant Recipients, the incidence of DGF following deceased-donor kidney transplantation (DDKT) is approximately 23%, while, for living related kidney transplantation (LRKT), it is around 2.8% [4]. In Thailand, where this study was conducted, the incidence of DGF following DDKT is approximately 28.9%, whereas, for LRKT, it is approximately 1.5% [5].

DGF has both short- and long-term adverse effects on KT outcomes. In the short term, patients with DGF require additional renal replacement therapy, which escalates hospitalization costs. Over the long term, DGF correlates with an augmented risk of acute rejection and diminished graft survival [6].

Various factors contribute to the occurrence of DGF. These factors encompass donor-related variables, such as the donor’s age, kidney size, expanded criteria donor (ECD) status, diabetes mellitus (DM) status, hypertension status, and history of stroke [7,8,9,10]. Graft handling factors, such as the cold ischemic time (CIT; e.g., CIT > 15 h has been associated with DGF) and the application of machine perfusion for donor kidney preservation pre-transplantation, also impact the occurrence of DGF [7,11]. In addition, recipient-related factors, including the recipient’s body size, DM status, the history of KT, blood transfusion history, and tissue histocompatibility, influence the occurrence of DGF [7,9,10].

Given the impact that DGF has on the outcomes of KT, efforts have been made to develop systems that can be used to preoperatively predict DGF risk, such as the DGF risk calculator [9], the DGFS score system [12], and the Jeldres scoring system [13]. However, the utilized models have been shown to exhibit limitations in terms of their sensitivity, positive predictive value (PPV), and negative predictive value (NPV) across different populations [9,12,13].

Recently, there has been mounting interest in leveraging machine learning (ML) techniques for predicting medical outcomes, and ML-based systems have demonstrated superior performance compared to traditional statistical analysis. Numerous studies have utilized clinical factors and various ML models, including random forest, extreme gradient boosting (XGBoost), and multilayer perceptron models for DGF prediction. Among these models, neural networks and multilayer perceptron models have exhibited relatively high accuracy in predicting DGF [14,15].

However, to date, no studies have amalgamated clinical data and histopathological data from post-implantation time-zero kidney biopsies for DGF risk prediction using ML techniques. Hence, in this study, we investigated whether integrating clinical data and histopathological data from post-implantation time-zero kidney biopsy samples with ML analysis can enhance the accuracy of DGF risk prediction. The primary objective was to assess the factors that correlated with DGF, and the secondary objective was to use traditional statistical and ML methods to construct equations based on clinical and histopathological factors that can be used to predict DGF in DDKT.

## 2. Materials and Methods

A retrospective cohort study was conducted at the Ramathibodi Excellence Center for Organ Transplantation, with data from January 2018 to December 2022. All adult DDKT recipients (aged 18 years or older) with ABO compatibility were considered eligible. All participants were required to have a negative T-cell and B-cell lymphocytotoxicity cross-match result. The exclusion criteria included the absence of time-zero kidney biopsy data, postoperative vascular or urological complications, a dual or en bloc kidney transplant, and incomplete data. Data on the following recipient variables were recorded: age, body mass index (BMI), underlying diseases, human leukocyte antigen (HLA) sensitization history (including blood transfusion and previous KT), HLA typing and mismatches, panel reactive antibodies (PRAs), residual renal function (RRF), and dialysis vintage. Similarly, data on the following donor variables were recorded: age, BMI, underlying diseases, terminal serum creatinine level, cause of death, donor type, acute kidney injury (AKI) stage (according to the Acute Kidney Injury Network [AKIN] criteria), and severity of proteinuria. Histopathological data from the post-implantation time-zero kidney biopsy samples were also recorded.

### 2.1. Sample Size

The sample size for this study was estimated using the formula shown below. Balaz et al. [16] found that patients who received kidneys from ECDs had a 54% incidence of DGF, while patients who received kidneys from standard criteria donors had a 34% incidence of DGF. When calculating the sample size, we took into account that the number of patients who received kidneys from brain-dead donors needed to be ≥220.
$$\begin{array}{c} n_{1}={\left[\frac{z_{1-\alpha /2}\sqrt{\bar{p\bar{q}}\left(1+\frac{1}{r}\right)}+z_{1-\beta }\sqrt{p_{1}q_{1}+\frac{p_{2}q_{2}}{r}}}{\Delta }\right]}^{2}\\ r=\frac{n_{2}}{n_{1}},\;q_{1}=1-p_{1},\;q_{2}=1-p_{2}\\ \bar{p}=\frac{p_{1}+rp_{2}}{1+r},\;\bar{q}=1-\bar{p} \end{array}$$
where n = sample size of the experimental group; P_1_ = proportion of the target population with the factors of interest compared to patients with the desired outcomes; P_2_ = proportion of the general population without the factors of interest compared to the reference group; Z_1−α_/2 = standard statistical value corresponding to the desired significance level (α) (Z_1−α_ = 1.96 for α = 0.05).

### 2.2. Clinical Characteristics and Outcomes

Data on baseline characteristics and outcomes were collected and compared between patients who developed DGF and patients who did not develop DGF. DGF was defined as the requirement for dialysis within one-week post-KT and was used as the outcome variable. 

### 2.3. Statistical Analysis

For the baseline characteristics, continuous variables were analyzed using Student’s t-test or Mann–Whitney U test, with the latter employed in the absence of normal data distribution. Categorical variables were analyzed using the χ^2^ test or Fisher’s exact test. Logistic regression analysis was conducted to investigate factors related to DGF. All data analyses were performed using STATA version 17.0.

The clinical prediction score was developed by assigning weights to coefficients of predictors in a multivariable logistic model. Model performance was reported using the following metrics: the area under the receiver operating characteristic curve (AUROC), accuracy, NPV, and PPV.

ML models were constructed utilizing clinical data, histopathological data from the time-zero kidney biopsy samples, and the Python programming language with various libraries, such as sci-kit learn, pandas, numpy, and keras. A nested cross-validation approach was employed to evaluate and optimize the ML models. The outer cross-validation loop was used to assess model performance, while the inner loop focused on hyperparameter tuning (Figure 1).

#### 2.3.1. Exploratory Data Analysis Conducted to Examine the Initial Data

Exploratory data analysis was performed to assess the data set’s initial structure, detect outliers, and evaluate data quality. Data on the following variables were missing, at the stated percentages: donor oliguria (0.28%), recipient hypertension (0.28%), arteriolar hyalinosis (Ah; 6.5%), and arterial intimal thickening (CV; 25.7%).

#### 2.3.2. Data Preprocessing

After a thorough chart review, data missing from the DGF prediction data set were determined to be missing completely at random. To address this, we applied the MICE methodology using the miceforest package. This process involves generating multiple imputations that account for uncertainty, analyzing the imputed data sets, and combining the results to ensure robust estimates. To prevent data leakage, the MICE methodology was applied exclusively to the training data set.

#### 2.3.3. Feature Engineering and Feature Selection

Feature engineering was applied to incorporate clinically relevant thresholds into the data set, enhancing the model’s interpretability and predictive capacity.

To select the features, we utilized the statistically significant variable and other variables that were manually selected by an expert nephrologist.

#### 2.3.4. Model Selection

The following three types of models were tested with hyperparameters optimized via GridSearchCV:Random forest: Tuned over n_estimators (100–400), max_depth (1–5), and min_samples_split (2–10);XGBoost: Optimized for n_estimators (100–500), max_depth (1–7), and learning_rate (0.001–0.3); andNeural network: Adjusted for various hidden_layer_sizes, activation functions (relu, tanh), and alpha (0.0001–0.01).

#### 2.3.5. Model Training and Evaluation

Nested cross-validation ensures robust model evaluation by using two loops: an inner loop for model selection and an outer loop for performance assessment. In this case, both loops used 10-fold CV. The inner loop optimized hyperparameters and selected the best model, which was then retrained on the full training set. The outer loop evaluated this model’s performance on unseen test data, using the F1 score to provide an unbiased estimate of generalization.

## 3. Results

Among the 517 patients initially identified for potential inclusion in this study, no time-zero kidney biopsy data were available for 138 (26.8%) patients. Of the remaining 379 patients, four (1%) underwent a dual kidney transplant, five (1.3%) experienced vascular and urinary complications, and 16 (4.2%) had incomplete data; thus, these patients were excluded from the analysis. Hence, 354 patients were included in this study (Figure 2).

Among the 354 patients included in this study, 64 (18.1%) experienced DGF, 86 (24.3%) exhibited slow graft function, and 206 (58.2%) demonstrated immediate graft function. The patients who experienced DGF are referred to herein as the DGF group, and those who experienced slow or immediate graft function are referred to as the non-DGF group.

### 3.1. Baseline Characteristics

Most of the recipients’ baseline characteristics were similar between the DGF and non-DGF groups. The exceptions were that the patients in the DGF group exhibited lower incidences of a history of HLA sensitization (PRA 42% vs. 59%, *p* = 0.014) and the cause of ESKD being immunoglobulin (Ig)A nephropathy (0% vs. 6.6%, *p* = 0.031) than those in the non-DGF group. Additionally, the DGF group had a higher proportion of males (73% vs. 57%, *p* = 0.017) and a higher number of patients with a BMI > 23 kg/m^2^ (52% vs. 38%, *p* = 0.050) compared to the non-DGF group. 

Most of the donors’ baseline characteristics were similar between the DGF and non-DGF groups, except for a higher prevalence of DM (9% vs. 3%, *p* = 0.025), a longer CIT (19 h vs. 17 h, *p* < 0.001), higher terminal serum creatinine levels (1.35 mg/dL vs. 1.01 mg/dL, *p* < 0.001), and a higher incidence of a history of AKI (86% vs. 67%) in the DGF group compared to the non-DGF group. A more severe donor AKI was observed in the DGF group compared to the non-DGF group (Table 1).

Regarding the time-zero kidney biopsy data, no differences were observed between the groups regarding any of the examined histopathological factors (i.e., the percentage of glomerulosclerosis, interstitial fibrosis/tubular atrophy [ci/ct] score, CV score, and Ah score; Table 1).

The median length of hospital stay was 15 days (interquartile range [IQR] 10–24 days). Patients with DGF had significantly longer hospital stays compared to those without DGF (30 days vs. 14 days, *p* < 0.001). At one month post-KT, the patients with DGF exhibited lower kidney function compared to those without DGF (estimated glomerular filtration rate: 36 ± 18 mL/min/1.73 m^2^ vs. 58 ± 21 mL/min/1.73 m^2^, *p* < 0.001).

### 3.2. Univariate and Multivariate Analyses of DGF Risk Factors

The results of the univariate and multivariate analyses of the DGF risk factors are shown in Table 2. The univariate analysis indicated that a donor BMI > 23 kg/m^2^, donor DM, a terminal serum creatinine level (per mg/dL), a CIT > 18 h, a donor AKIN score of 2–3, a male recipient, and a ci/ct score of 2–3 in the time-zero kidney biopsy sample all contributed to the occurrence of DGF. 

The multivariate analysis revealed that the following variables were independently predictive factors of DGF: a donor BMI > 23 kg/m^2^ (odd ratio [OR]: 2.17, 95% CI: [1.16, 4.05]; *p* = 0.015), donor DM (OR: 3.72, 95% CI: [1.17, 11.80]; *p* = 0.026), a CIT > 18 h (OR: 2.35, 95% CI: [1.26, 4.37]; *p* = 0.007), a male recipient (OR: 2.38, 95% CI: [1.18, 4.78]; *p* = 0.015), and a ci/ct score of 2–3 in the time-zero kidney biopsy sample (OR: 6.42, 95% CI: [1.73, 23.87]; *p* = 0.015).

### 3.3. Prediction of DGF in DDKT Using the Traditional Statistical Method and Clinical Factors

Many clinical centers do not have the capability to perform time-zero kidney biopsies. Therefore, we worked to develop an equation that can be used to predict DGF on the basis of clinical factors only. We undertook a multivariable binary logistic analysis and employed a backward stepwise regression to construct the equation, which is shown below
$$y=\frac{e^{0.6929168\cdot BMI_{D\;group\;23}+1.368313\cdot DM_{D}-0.9165385\cdot CIT_{D\;group\;18}+0.5376716\cdot TerminalCr+0.7113935\cdot Sex_{R}+1.78028\cdot CI_{group\;1}-4.003639}}{1+e^{0.6929168\cdot BMI_{D\;group\;23}+1.368313\cdot DM_{D}-0.9165385\cdot CIT_{D\;group\;18}+0.5376716\cdot TerminalCr+0.7113935\cdot Sex_{R}+1.78028\cdot CI_{group\;1}-4.003639}}$$

Scoring details: BMI donor ≥ 23 kg/m² (score = 1), < 23 kg/m² (score = 0)

Cold ischemic time > 18 h (score = 1), ≤ 18 h (score = 0)

Donor DM (score = 1), Male recipient (score = 1)

Terminal Cr (mg/dL)

In this equation, y denotes the probability of DGF, and its value ranges from 0 to 1. As shown in Figure 3, an AUROC of 0.750 (95% CI: [0.687, 0.813]) was obtained using this equation.

Subsequently, the equation was applied in a practical setting to determine its predictive performance. With a probability score of 0.273, it was observed that the equation could predict the occurrence of DGF in DDKT patients with a specificity of 84.48% and a sensitivity of 46.88%. The positive likelihood ratio was 3.02, and the negative likelihood ratio was 0.629 (Table 3).

### 3.4. Prediction of DGF in DDKT Using the Traditional Statistical Method and Clinical and Histopathological Factors

We performed a multivariable binary logistic analysis and used a backward stepwise regression to develop an equation that can be used to predict the probability of DGF. The equation is shown below.
$$y=\frac{e^{0.7669392\cdot BMI_{D\;group\;23}+1.456151\cdot DM_{D}-0.8935933\cdot CIT_{D\;group\;18}+0.5541554\cdot TerminalCr+0.9249727\cdot Sex_{R}+1.78028\cdot CI_{group\;1}-4.003639}}{1+e^{0.7669392\cdot BMI_{D\;group\;23}+1.456151\cdot DM_{D}-0.8935933\cdot CIT_{D\;group\;18}+0.5541554\cdot TerminalCr+0.9249727\cdot Sex_{R}+1.78028\cdot CI_{group\;1}-4.003639}}$$

Scoring details: BMI donor ≥ 23 kg/m² (score = 1), < 23 kg/m² (score = 0)

Cold ischemic time > 18 h (score = 1), ≤ 18 h (score = 0)

Donor DM (score = 1), Male recipient (score = 1)

Terminal Cr (mg/dL), CI score 2–3 (score = 1)

An AUROC of 0.769 (95% CI: [0.707, 0.831]) was achieved using this equation, and its performance was compared with that of the equation based on clinical factors only in Figure 3. The equation was then applied in a practical setting, and, with a probability threshold of 0.265, the equation demonstrated a specificity of 83.10% and a sensitivity of 56.25%. The positive likelihood ratio was 3.33, and the negative likelihood ratio was 0.53 (Table 3).

### 3.5. Performance of the ML Models in Predicting DGF in DDKT

The variables included in the ML models were selected on the basis of expert opinion and the multivariable logistic regression results. In terms of the donor-related factors incorporated into the models, the terminal creatinine level and urine protein–creatinine ratio (UPCR) were included because they were identified by the nephrologists as critical variables. The donor BMI and DM status, as well as the CIT, were included based on their significance in the multivariable analysis. Regarding the recipient-related factors, recipient sex was included based on its potential influence on graft function. The time-zero kidney biopsy parameters ci/ct and Ah were also included, with Ah status being identified as a key factor by the nephrologists. These variables were utilized as inputs for the ML models to leverage both clinical expertise and statistical rigor and thus enhance the predictive performance of the models.

The predictive performances of the ML models in terms of their ability to accurately determine the probability of DGF occurring in DDKT patients are summarized in Figure 4 and Table 4.

The random forest model was found to have a sensitivity of 22.7%, a specificity of 100%, a PPV of 89.1%, and an NPV of 85.5%. The neural network model was found to have a sensitivity of 33.3%, a specificity of 97.4%, a PPV of 66.0%, and an NPV of 87.0%. The XGBoost model was found to have a sensitivity of 89.1%, a specificity of 99.8%, a PPV of 98.4%, and an NPV of 97.8%. The nested cross-validation results for the prediction of DGF in DDKT cases using the random forest, neural network, and XGBoost models are shown in Table 5.

The XGBoost model achieved the highest F1 score of 1, which indicated perfect classification. This was likely because of this model’s ability to handle feature interactions with a deep structure (300 estimators). The neural network model achieved the next highest F1 score (0.7551) by leveraging multiple hidden layers and the tanh activation function. In contrast, the random forest model, with its shallow depth and default splitting parameters, produced the lowest F1 score (0.48).

## 4. Discussion

In this retrospective single-center study, we identified factors that influenced the occurrence of DGF in kidney transplant recipients (KTRs), including a donor BMI > 23 kg/m^2^, donor DM, a CIT > 18 h, a male recipient, and a ci/ct score of 2–3 in the time-zero kidney biopsy sample. Various equations have been developed to predict the occurrence of DGF [9,12,13]. However, a common limitation of these models is their poor performance across diverse patient populations. This is mainly because they rely solely on clinical factors.

It has been demonstrated that pathological abnormalities in tissue samples obtained from pre-implantation kidney biopsies are associated with DGF in KTRs [16,17]. To date, no equation has integrated both clinical factors and pathological findings in post-implantation time-zero kidney biopsy samples to forecast DGF in KTRs. Hence, we aimed to devise a predictive equation for DGF in KTRs based on clinical and histopathological factors, utilizing ML techniques. These contemporary methods aid in enhancing the performance of predictive equations for DGF in KTRs [15].

Interestingly, our findings diverged from those of previous studies in certain aspects. In contrast to others [11], we did not observe a significant association between DGF and recipients with hypertension, DM, and heart disease. This disparity might be attributed to differences in patient demographics; our study included fewer recipients with DM and heart disease (12–13%) than other studies that had higher prevalences (24% and 38%) [9,12].

Furthermore, this is the first study to report a significant association (i.e., a positive correlation) between the donor’s DM status and DGF. Our center, like many across Thailand, utilizes stricter donor selection criteria than those in other regions that accept donors with various underlying conditions. Hence, our selection practices may have impacted the correlation between donor DM and DGF incidence. This aspect of our study highlights the influence that local medical practices and demographic variations have on the outcomes of KT, underscoring the need for regional considerations in predictive model development and clinical decision making.

In our study, we did not find a significant association between high PRA values or HLA mismatches and DGF. This is consistent with other studies [12] but in contrast to the results of other studies [9,13]. This might have been because of the predominantly low immunological risk in our patient cohort, which was evident from the median PRA value being 0 and the absence of patients with more than four HLA mismatches.

The proportion of ECDs in our study (18.4%) was within the range reported in the literature (13%–40%) [9,12]. However, contrary to the results of others that have suggested that ECDs elevate the risk of DGF by 1.7 times [18], our results did not indicate a relationship between ECDs and DGF. In addition, pathological factors conventionally associated with DGF, such as CV, Ah, and glomerulosclerosis > 20%, were not linked to DGF in our study. We speculate that this could be attributed to our reliance on wedge kidney biopsies, which typically yield fewer vessels compared to core biopsies. Consequently, only 255 of the 354 donated kidneys had pathology results pertinent to two blood vessels. The median number of glomeruli that showed sclerosis was 4 (IQR: 0–13), with a significant proportion of kidneys demonstrating > 20% glomerulosclerosis related to DGF.

Regarding the performance of the equations we developed in this study, our equation that integrated clinical and histopathological data resulted in a higher AUROC value (0.78) than prior equations. In 2017, Michalak et al. [19] analyzed various DGF prediction equations, such as those proposed by Irish et al., Chapal et al., and Jeldres et al., and reported that the associated AUROC values ranged from 0.52 to 0.69. 

When we evaluated the performance of three ML models (random forest, neural network, and XGBoost models) in predicting DGF in DDKT, we found that each model exhibited distinct advantages. This indicated that the clinical priorities must be considered in each case during the model selection process.

The random forest model excelled in specificity (100%) and NPV (85.5%), meaning it was effective in identifying patients unlikely to develop DGF. However, its sensitivity was only 22.7%, indicating an increased risk of missing high-risk patients. The neural network model demonstrated a more balanced performance, with a sensitivity of 33.34% and specificity of 97.4%. Its accuracy (85.8%) and AUC (0.8445) suggested a reasonable ability to identify both at-risk and non-at-risk patients. However, its Brier score (0.102) indicated room for improvement in probability estimates. The XGBoost model stood out for its high sensitivity (89.1%) and best overall accuracy (97.9%), making it the top performer in detecting at-risk patients. Its low Brier score (0.020) also indicated reliable probability estimates. Despite its high sensitivity, the wide confidence intervals suggest variability in performance across data sets.

Based on our results, XGBoost is the most suitable model for predicting DGF because of its high sensitivity and accuracy, making it particularly effective for identifying at-risk patients. XGBoost incorporates L1 (Lasso) and L2 (Ridge) regularization, which penalize complex models. This is crucial in medical data sets where overfitting can occur due to noisy data or a high number of features relative to the number of observations. Its strong ability to differentiate between patients who will and will not develop DGF, combined with reliable probability estimates, supports its use in clinical scenarios where early detection and intervention are critical.

It is not possible to directly compare the performances of traditional statistical models and ML models, as traditional statistical models do not have a test set. Therefore, external validation should be performed on another test set for comparison.

The models varied in their emphasis on factors such as warm ischemia time (WIT) and donor terminal creatinine, indicating model-specific sensitivities and potential interactions not uniformly captured. This highlights the need for the cautious interpretation and consideration of both data insights and clinical expertise. While the predictive value of WIT varied, its biological impact on organ viability is clear. The models also identified donor urine volume and the AKI stage as important factors, warranting further clinical investigation. Additionally, the predictive value of capillary wall wrinkling, an emerging marker, reflects the evolving nature of transplant pathology. Our study underscores the potential of ML to augment clinical decision making, although it also highlights the need for meticulous validation and the integration of model findings with established clinical frameworks.

This study has several strengths, which are primarily underscored by the novel integration of both clinical data and histopathological data from post-implantation time-zero kidney biopsies to develop a predictive model for DGF in DDKT patients. This is the first report of ML methodologies being applied in this field, and utilizing this approach allowed us to overcome some of the limitations inherent in traditional statistical methods. Utilizing advanced ML techniques also enhanced the capability to discern complex patterns within the data, which improved the predictive accuracy of the tested models. Another strength of this study is that the performance of the models was comprehensively evaluated via the use of key metrics (i.e., AUROC, sensitivity, specificity, and predictive values). Thus, the models were thoroughly assessed, and the results provided critical information relevant to the potential clinical application of such models. This research marks a considerable advancement in transplantation medicine, contributing substantially to the scientific literature and laying a robust foundation for future investigations into predictive modeling and personalized patient care in KT.

Incorporating clinical and histopathological factors into machine learning models, such as our XGBoost-based approach for predicting delayed graft function (DGF), mirrors advancements in healthcare applications of AI. Deep learning models, like the Deep Ensemble Adaptive Architectures for COVID-19 detection, have demonstrated high precision and adaptability in complex settings [20]. CNN-based frameworks used in Alzheimer’s [21] and brain tumor [22] diagnostics emphasize the transformative impact of integrating multi-source data and highlight the potential for ML to refine predictive accuracy, enhance early detection, and support personalized interventions in clinical practice.

It is important to note that this study has some limitations. First, data were collected on deceased kidney donors. Second, the sample size of the time-zero kidney biopsy was small (n = 354), which may have reduced the effectiveness of the equations used to predict DGF. Despite this constraint, the sample size calculation was meticulously performed based on the existing literature and indicated that our sample size exceeded the minimum requirement. Another limitation was the inadequacy of the time-zero kidney biopsy samples. A quarter (25%) of the samples did not include blood vessels because of the biopsy technique used, which resulted in relatively less tissue. Therefore, using a different biopsy technique may improve the predictive performance of the models. Furthermore, retrospective data from a single center were used in this study. Thus, further studies are required to externally validate the results in diverse patient populations across different centers to ascertain the generalizability and applicability of the model.

## 5. Conclusions

In summary, we have developed an ML-based model for predicting DGF that has a high performance compared to models developed using traditional approaches. The XGBoost model emerged as the most effective among the tested ML models. This novel research represents a significant advancement in personalized medicine, and the developed model could be used by clinicians to make informed decisions about kidney acceptance and the optimal timing of biopsy for the cause of early allograft dysfunction to reduce the risk of procedure and optimize outcomes. Such models may be particularly useful in resource-limited settings. To ensure the generalizability and effectiveness of this model in real-world scenarios, external validation of the results of this study is imperative.

## Figures and Tables

**Figure 1 jcm-13-07502-f001:**
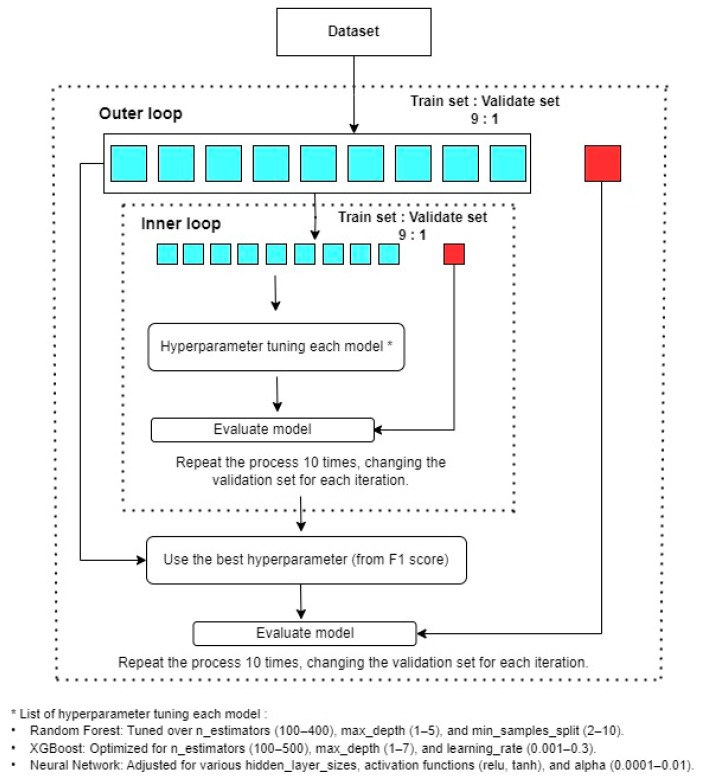
Illustration of the nested cross-validation process for hyperparameter tuning and model evaluation. The outer loop managed data partitioning for validation, while the inner loop optimized hyperparameters. The models were evaluated using the best hyperparameters based on the F1 scores, and repeated iterations were performed to ensure a robust performance assessment. The red box in the outer loop represents the validation set used for final model evaluation, while the red box in the inner loop represents the validation set used for hyperparameter tuning.

**Figure 2 jcm-13-07502-f002:**
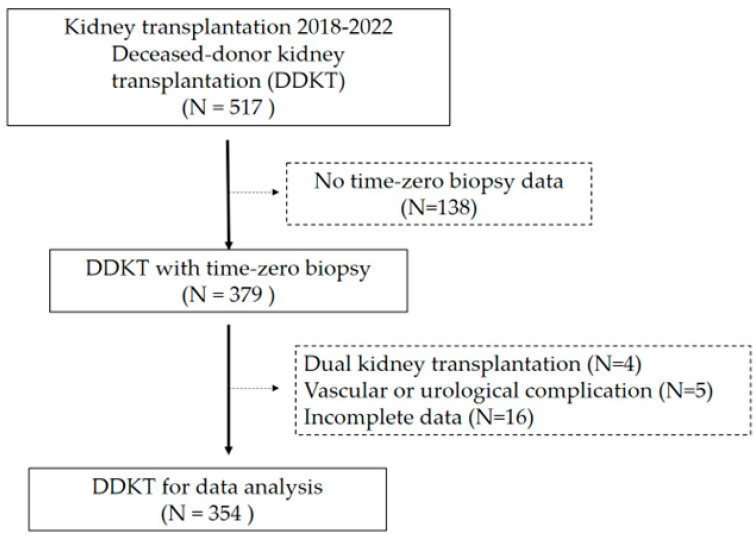
Study flow diagram.

**Figure 3 jcm-13-07502-f003:**
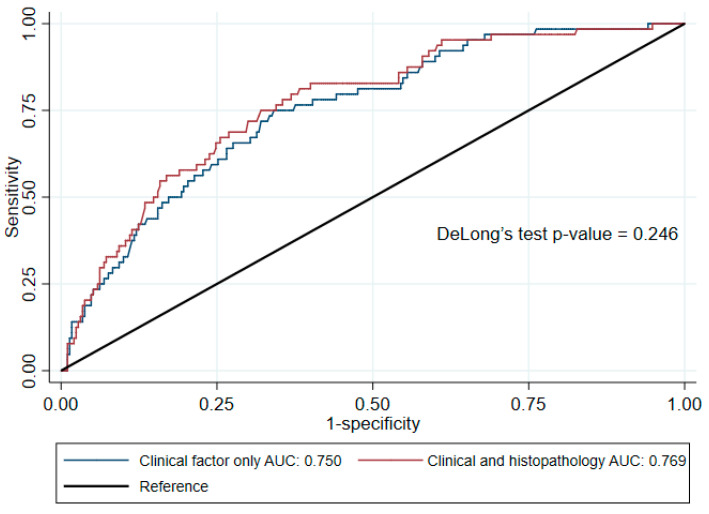
The receiver operating characteristic curves obtained using the equations developed to predict the probability of DGF in deceased-donor kidney transplantation (DDKT) patients using clinical factors only (blue) and using clinical and histopathological factors (red). DeLong’s test did not show a statistically significant difference between the AUCs of the two models (*p* = 0.246).

**Figure 4 jcm-13-07502-f004:**
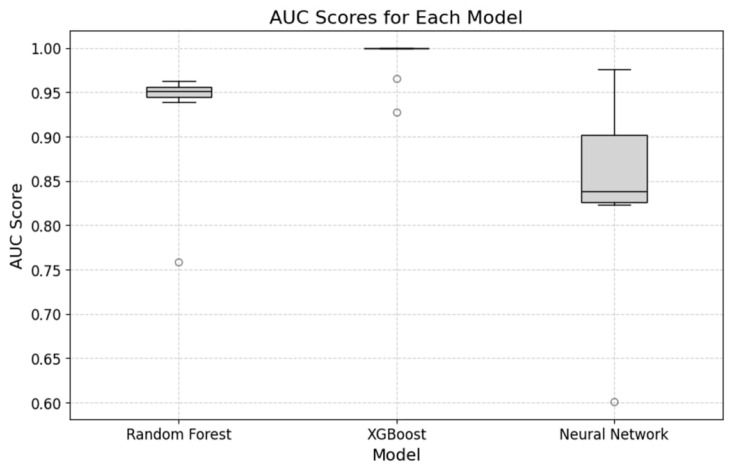
AUC score distribution for three models (random forest, XGBoost, and neural network) used to predict delayed graft function in deceased donor kidney transplants. Each boxplot illustrates the variability and median AUC across cross-validation, highlighting the performance consistency of each model.

**Table 1 jcm-13-07502-t001:** Analysis of the baseline characteristics of the patients in the delayed graft function (DGF) and non-DGF groups.

Characteristic	Total (n = 354)	Non-DGF (n = 290)	DGF (n = 64)	*p*
Recipient				
Age, mean ± SD years	44 ± 11	43 ± 10	44 ± 12	0.523
Male, n (%)	213 (60)	166 (57)	47 (73)	0.017 *
BMI ≥ 23 kg/m^2^, n (%)	144 (41)	111 (38)	33 (52)	0.050
Underlying disease, n (%)				
Hypertension	299 (85)	246 (85)	53 (83)	0.695
DM	45 (13)	35 (12)	10 (16)	0.440
ASCVD	41 (12)	34 (12)	7 (11)	0.859
Smoking, n (%)	22 (6)	18 (6)	4 (6)	1.000
Sensitization history, n (%)	198 (56)	171 (59)	27 (42)	0.014 *
HLA mismatches, n (%)				
0–4	354 (100)	290 (100)	64 (100)	
5–6	0	0	0	1.000
Any DR mismatch, n (%)	118 (33.3)	98 (33.7)	20 (31.2)	0.183
PRA, median (IQR)	0 (0–0)	0 (0–0)	0 (0–0)	0.723
Cause of ESKD, n (%)				
Unknown	255 (72)	203 (70)	52 (81)	0.070
Diabetic nephropathy	21 (6)	18 (6)	3 (5)	0.778
ADPKD	5 (1)	4 (1)	1 (2)	1.000
IgA nephropathy	19 (5)	19 (7)	0 (0)	0.031 *
Lupus nephritis	14 (4)	13 (5)	1 (2)	0.480
Chronic glomerulopathy	22 (6)	18 (6)	4 (6)	1.000
RRF, n (%)				
<300 mL/day	283 (80)	228 (79)	55 (86)	
301–500 mL/day	38 (11)	33 (11)	5 (8)	
>501 mL/day	33 (9)	29 (10)	4 (6)	0.414
Dialysis mode HD, n (%)	302 (85)	243 (84)	59 (92)	0.076
Dialysis vintage, median (IQR) years	5 (3–8)	5 (3–8)	5 (3–8)	0.813
Induction regimen, n (%)				
ATG	48 (13)	40 (14)	8 (13)	
Anti-IL-2R	264 (75)	217 (75)	47 (73)	0.820
Donor				
Age, mean ± SD years	40 ± 14	40 ± 14	39 ± 13	0.6008
Male, n (%)	274 (77)	221 (76)	53 (83)	0.253
BMI ≥ 23 kg/m^2^, n (%)	194 (55)	150 (52)	44 (69)	0.013 *
Underlying disease, n (%)				
Hypertension	59 (17)	44 (15)	15 (23)	0.108
DM	14 (4)	8 (3)	6 (9)	0.025 *
ASCVD	11 (3)	10 (3)	1 (2)	0.136
ECD, n (%)	65 (18)	53 (18)	12 (19)	0.929
History of cardiac arrest, n (%)	42 (12)	37 (13)	5 (8)	0.268
Cause of death, n (%)				
CVA/Stroke	121 (34)	94 (32)	27 (42)	0.136
Head trauma	221 (62)	185 (64)	36 (56)	0.217
Smoking, n (%)	193 (55)	158 (54)	35 (55)	0.936
CIT, mean ± SD h	18 ± 4	17 ± 4	19 ± 4	<0.001 *
WIT, mean ± SD min	50 ± 20	50 ± 20	52 ± 21	0.432
Urine volume, median (IQR) mL/kg/h	2.3 (1.4–3.9)	2.3 (1.3–3.7)	2.3 (1.7–4.4)	0.482
Creatinine, median (IQR) mg/dL				
Initial	0.83 (0.67–1.00)	0.82(0.67–0.99)	0.88 (0.73–1.08)	0.493
Terminal	1.11 (0.79–1.48)	1.01 (0.77–1.40)	1.35 (1.07–1.91)	<0.001 *
Peak	1.30 (0.99–1.83)	1.23 (0.95–1.72)	1.52 (1.22–2.18)	0.003 *
AKI stage, n (%)				
NO	105 (30)	96 (33)	9 (14)	
AKIN 1	130 (37)	106 (37)	24 (38)	
AKIN 2	82 (23)	62 (21)	20 (31)	
AKIN 3	37 (10)	26 (9)	11 (17)	0.007 *
Proteinuria UPCR, n (%)				
<0.3 g/d	119 (33)	98 (34)	24 (38)	
0.3–1 g/d	140 (40)	117 (40)	20 (31)	
1–3 g/d	85 (24)	69 (24)	16 (25)	
>3 g/d	10 (3)	6 (2)	4 (6)	0.282
Time-zero kidney biopsy				
Glomerulosclerosis, median (IQR)	4 (0–13)	4 (0–13)	2 (0–11)	0.7179
Ci/Ct score, n (%)				
0	123 (35)	101 (35)	22 (34)	
1	216 (61)	180 (62)	36 (56)	
2	13 (4)	8 (3)	5 (8)	
3	2 (1)	1 (0)	1 (2)	0.128
CV score, n (%)				
Unknown	91 (26)	78 (27)	13 (20)	
0	197 (55)	161 (56)	36 (56)	
1	54 (15)	42 (14)	12 (19)	
2	9 (3)	6 (2)	3 (5)	
3	3 (1)	3 (1)	0 (0)	0.503
Ah score, n (%)				
Unknown	23 (6)	33 (11)	1 (2)	
0	212 (60)	175 (60)	37 (58)	
1	108 (31)	82 (28)	26 (40)	
2	11 (3)	0 (0)	0 (0)	
3	0 (0)	0 (0)	0 (0)	0.096
Capillary wall wrinkling				
No	312 (88)	259 (89)	53 (83)	
Yes	42 (12)	31 (11)	11 (17)	0.146

ADPKD, autosomal dominant polycystic kidney disease; Ah, arteriolar hyalinosis; AKI, acute kidney injury; AKIN, Acute Kidney Injury Network; Anti-IL-2R, anti-interleukin-2 receptor monoclonal antibody; ASCVD, atherosclerotic cardiovascular disease; ATG, anti-thymocyte globulin; BMI, body mass index; CIT, cold ischemic time; Ci/Ct, interstitial fibrosis/tubular atrophy; CV, arterial intimal thickening; CVA, cerebral vascular accident; DM, diabetes mellitus; ECD, expanded criteria donor; ESKD, end-stage kidney disease; HD, hemodialysis; HLA, human leukocyte antigen; IgA, immunoglobulin A; IQR, interquartile range; PRA, panel reactive antibody; RRF, residual renal function; SD, standard deviation; UPCR, urine protein–creatinine ratio; WIT, warm ischemic time; and * *p* value < 0.05, statistical significance.

**Table 2 jcm-13-07502-t002:** Univariate and multivariate analyses of DGF risk factors.

Parameter	Univariate OR (95% CI)	*p*	Multivariate OR (95% CI)	*p*
**Donor**				
BMI ≥ 23 kg/m^2^	2.05 (1.15, 3.65)	0.014	2.17 (1.16, 4.05)	0.015 *
DM	3.65 (1.22, 10.91)	0.020	3.72 (1.17, 11.80)	0.026 *
Terminal creatinine, per mg/dL	1.80 (1.28, 2.52)	0.001	1.48 (0.98, 2.22)	0.063
CIT > 18 h	2.43 (1.39, 4.24)	0.002	2.35 (1.26, 4.37)	0.007 *
AKIN score of 2–3	2.14 (1.24, 3.72)	0.007	1.65 (0.76, 3.59)	0.206
UPCR > 1	3.30 (0.89, 12.21)	0.073	3.42 (0.71, 16.63)	0.126
ECD	1.03 (0.52, 2.07)	0.929		
**Recipient**				
Male	2.07 (1.13, 3.77)	0.018	2.38 (1.18, 4.78)	0.015 *
BMI > 23 kg/m^2^	1.72 (0.99, 2.96)	0.052	1.39 (0.47, 3.21)	0.311
Age	1.01 (0.98, 1.03)	0.521		
PRA %	0.99 (0.99, 1.01)	0.722		
DM	1.33 (0.63, 2.89)	0.441		
**Histopathological**				
Ci/Ct score of 2–3	3.23 (1.11, 9.42)	0.032	6.42 (1.73, 23.87)	0.006 *
Ah+	1.32 (0.75, 2.32)	0.332		
CV score of 2–3	1.41 (0.37, 5.40)	0.621		
Glomerulosclerosis > 15%	0.96 (0.40, 2.27)	0.920		
Capillary wall wrinkling	1.73 (0.82, 3.66)	0.151		

Ah, arteriolar hyalinosis; AKIN, Acute Kidney Injury Network; BMI, body mass index; CI, confidence interval; CIT, cold ischemic time; Ci/Ct, interstitial fibrosis/tubular atrophy; CV, arterial intimal thickening; DM, diabetes mellitus; ECD, expanded criteria donor; OR, odd ratio; PRA, panel reactive antibody; and UPCR, urine protein–creatinine ratio. * *p* value < 0.05, statistical significance.

**Table 3 jcm-13-07502-t003:** Performance of the equations developed using a traditional statistical method.

	Probability	Sensitivity	Specificity	AUC	LR+	LR−
**Clinical and histopathological factors**	0.265	0.562	0.831	0.769	3.330	0.530
**Clinical factors only**	0.273	0.468	0.844	0.750	3.020	0.629

AUC, area under the curve; LR+, positive likelihood ratio; LR−, negative likelihood ratio.

**Table 4 jcm-13-07502-t004:** Performance of the ML models in terms of their ability to predict the occurrence of DGF in DDKT patients.

**Model**	**Sensitivity**	**Specificity**	**Accuracy**	**AUC**	**PPV**	**NPV**	**Brier Score**
**Random forest**	0.227 (0.169, 0.273)	1.000 (0.999, 1.000)	0.860 (0.849, 0.868)	0.932 (0.892, 0.955)	0.891 (0.691, 1.000)	0.855 (0.846, 0.861)	0.097 (0.092, 0.107)
**Neural network**	0.333 (0.230, 0.433)	0.974 (0.966, 0.983)	0.858 (0.844, 0.876)	0.845 (0.780, 0.896)	0.660 (0.498, 0.769)	0.870 (0.853, 0.887)	0.102 (0.087, 0.115)
**XGBoost**	0.891 (0.736, 1.000)	0.998 (0.995, 1.000)	0.979 (0.949, 1.000)	0.989 (0.972, 1.000)	0.984 (0.955, 1.000)	0.978 (0.947, 1.000)	0.020 (0.004, 0.041)

AUC, area under the curve; NPV, negative predictive value; PPV, positive predictive value; and XGBoost, extreme gradient boosting.

**Table 5 jcm-13-07502-t005:** F1 scores and hyperparameters from the nested cross-validation for predicting DGF using the random forest, neural network, and XGBoost models.

Model	Best F1 Score	Parameters
Random forest	0.48	Max depth = 5, minimum samples split = 2, number of estimators = 100
Neural network	0.7551	Activation = tanh, alpha = 0.001, hidden layer size = (20, 100, 100, 20)
XGBoost	1	Learning rate = 0.1, max depth = 5, number of estimators = 300

## Data Availability

The data that support the findings of this study are available upon request from the corresponding author. The data are not publicly available due to privacy and ethical restrictions.

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
