# Peer review of "Integrating Clinical and Histopathological Data to Predict Delayed Graft Function in Kidney Transplant Recipients Using Machine Learning Techniques"

_jcm, 2024, doi:10.3390/jcm13247502_

Round 1

Reviewer 1 Report

Comments and Suggestions for Authors

Dear authors,

I appreciate your first study to combine clinical and histopathological data for DGF prediction. Novel application of multiple machine learning approaches, and comprehensive comparison of different modeling approaches was very required in this field.

Also, the articles has clear potential clinical application, and identification of novel predictive factors. This would have practical implications for resource-limited settings.

However, I think it still needs some major revisions:

1. Comparison to the "Deep Learning" research is needed in discussion section, especially ones that discussed multi-modal medical data integration, similar to your paper combines clinical and histopathological data. This could provide insights into handling multiple types of medical imaging data in ML models. Example of such ones are Recent Deep Learning-Based Brain Tumor Segmentation Models Using Multi-Modality Magnetic Resonance Imaging: A Prospective Survey; Enhancing Alzheimer's Disease Diagnosis and Staging: A Multistage CNN Framework Using MRI; An Adaptive Ensemble Deep Learning Framework for Reliable Detection of Pandemic Patients; etc... 

2. The extremely high performance metrics (e.g., XGBoost with 0.989 AUROC) suggest potential overfitting. Meanwhile, wide confidence intervals in the XGBoost results indicate model instability. I am curious authors could do additional external validation. If you cannot, kindly give us clear explanation of how the training/test split was performed. Also, expanded description of feature engineering and preprocessing steps is required.

3. In the cohort, there is lower prevalence of diabetes and heart disease compared to other studies, and I found predominantly low immunological risk patients (median PRA of 0). Some readers may think results may not generalize to centers with different donor selection criteria. Could you make a defense on this?

Minor revisions:

1. Missing Informations. for example, No clear description of handling missing data in the validation set, limited details about model hyperparameter selection process, and incomplete reporting of confidence intervals for some metrics

2. Presentation quality. Some figures lack clear legends and explanations, and I see inconsistent reporting of performance metrics across different models.

Thank you for your valuable contributions to our field of research. I look forward to receiving the revised manuscript.

Author Response

  1. 1. Comparison to the "Deep Learning" research is needed in discussion section, especially ones that discussed multi-modal medical data integration, similar to your paper combines clinical and histopathological data. This could provide insights into handling multiple types of medical imaging data in ML models. Example of such ones are Recent Deep Learning-Based Brain Tumor Segmentation Models Using Multi-Modality Magnetic Resonance Imaging: A Prospective Survey; Enhancing Alzheimer's Disease Diagnosis and Staging: A Multistage CNN Framework Using MRI; An Adaptive Ensemble Deep Learning Framework for Reliable Detection of Pandemic Patients; etc...

 (LINE 484-491) and add reference 20-22

Abidin, Z.U.; Naqvi, R.A.; Haider, A; Kim, H.S.; Jeong, D; Lee, S.W. Recent deep learning-based brain tumor segmentation models using multi-modality magnetic resonance imaging: a prospective survey. Front Bioeng Biotechnol 2024, 12,1392807.

Ali, M.U.; Kim K.S.; Khalid, M.; Farrash, M; Zafar, A.; Lee, S.W. Enhancing Alzheimer's disease diagnosis and staging: a multistage CNN framework using MRI. Front Psychiatry 2024, 15, 1395563.

Iqbal, M.S.; Naqvi, R.A.; Alizadehsani, R.; Hussain, S.; Moqurrab, S.A.; Lee, S.W.  An adaptive ensemble deep learning framework for reliable detection of pandemic patients. Comput Biol Med 2024, 168,107836.

  1. 2. The extremely high performance metrics (e.g., XGBoost with 0.989 AUROC) suggest potential overfitting. Meanwhile, wide confidence intervals in the XGBoost results indicate model instability. I am curious authors could do additional external validation. If you cannot, kindly give us clear explanation of how the training/test split was performed. Also, expanded description of feature engineering and preprocessing steps is required.

We currently don’t have external validation dataset yet. To mitigate overfitting and ensure robust model evaluation, we used a nested cross-validation approach for training and testing. Specifically, the outer loop of the nested cross-validation handled the division of data into train and test splits, while the inner loop focused on hyperparameter tuning through an additional layer of cross-validation. This method allowed us to optimize the model parameters without contaminating the test set, effectively reducing the risk of overfitting. (LINE 180-183, Figure 2 ),LINE 450-452

In the outer loop, we divided the data into multiple folds, iterating through each fold to ensure every data point had a chance to be part of the test set. Within each training fold, the inner loop further split the data to tune hyperparameters for each model, with the optimal parameters then applied to the test fold of the outer loop. By structuring the cross-validation in this nested manner, we achieved more reliable performance metrics and mitigated model instability, as it helped prevent data leakage between training and testing stages. (LINE 180-183, Figure 2 )

  1. 3. In the cohort, there is lower prevalence of diabetes and heart disease compared to other studies, and I found predominantly low immunological risk patients (median PRA of 0). Some readers may think results may not generalize to centers with different donor selection criteria. Could you make a defense on this?

In our study, we did not find a significant association between high PRA values or HLA mismatches and DGF. This is consistent with other studies [12] but in contrast to the results of other studies [9,13]. This might have been because of the predominantly low immunological risk in our patient cohort, which was evident from the median PRA value being 0 and the absence of patients with more than four HLA mismatches. (LINE 412-416}

The data from this study aligns with the data from Thailand, as shown in the table.

PRA

Non-DGF (N=290)

DGF(N=64)

0

233

52

1-10

4

2

11-20

2

0

21-30

6

1

31-40

3

2

41-50

3

1

51-60

7

0

61-70

4

0

71-80

7

1

81-90

11

2

91-100

10

3

Data from the Thai Transplant Society shows that in 2022, 80.6% of patients who received kidney transplants had a PRA of 0, while 8.9% had a PRA between 1-49%, 5.6% had a PRA between 50-80%, and 5% had a PRA between 81-100%.

Due to the limited number of kidney donors in Thailand, with only about 500 deceased donors available for kidney transplants each year, patients must be carefully assessed for suitability. Patients with diabetes may have complications, such as a history of stroke (old CVA) or coronary artery disease (CAD), which have not been adequately treated. This makes it less likely for these patients to receive a kidney transplant. Registry Book 2022 (Thai version).pdf - Google ไดรฟ์

 Minor revisions:

  1. 1. Missing Informations. for example, No clear description of handling missing data in the validation set, limited details about model hyperparameter selection process, and incomplete reporting of confidence intervals for some metrics

In our study, missing data were addressed using Multiple Imputation by Chained Equations (MICE), which we applied to the training set exclusively to prevent data leakage. This approach allowed us to create multiple imputed datasets. For the validation set, we used the same imputed values obtained from the training set's MICE model, ensuring consistency and avoiding potential bias. We have now clarified this process in the methods section. (Line198-200 )

Hyperparameters for each model were selected using a grid search approach within a nested cross-validation framework, with an inner 10-fold cross-validation to tune parameters on each training split. (LINE 202-212)

  1. 2. Presentation quality. Some figures lack clear legends and explanations, and I see inconsistent reporting of performance metrics across different models.

We improved the quality of the figures 2, 5 and 7.

Reviewer 2 Report

Comments and Suggestions for Authors

1.     The choice of only Random Forest and Neural Network for model comparison raises questions. Please justify this selection and discuss why other models were not included in the analysis.

2.     XGBoost is indicated as the model with the best performance. Please elaborate on why this model achieved such results and how it fits the characteristics of your dataset. Discuss the specific features of XGBoost that may have contributed to its success.

3.     Table 1 shows significant differences in sex and BMI among recipient data. Please clarify how these differences were addressed in your analysis. Discuss whether these factors were controlled for in the models and how they might influence the outcomes.

4.     In Figure 5, the AUC results appear to be similar whether or not biopsy data is included. Please provide a detailed analysis of these results. Are there any statistically significant differences between the two approaches? A clearer explanation would help contextualize the findings.

5.     The calculation of the LR+ (likelihood ratio positive) in Table 3 needs further clarification. Explain the formula used and why the values are greater than 1.

6.     Figure 7 lacks information regarding the number of tests conducted for each model. Please include this information to provide a clearer understanding of the robustness and reliability of the results.

7.     More related advances (VIEW 2023, 4, 20220039; VIEW 2023, 4, 20220038) should be included for discussion.

8.     Image Clarity: The clarity of the images (e.g., Figure 2) needs to be improved.

Comments on the Quality of English Language

/

Author Response

  1. 1.The choice of only Random Forest and Neural Network for model comparison raises questions. Please justify this selection and discuss why other models were not included in the analysis.

Our decision to focus on Random Forest and Neural Network models was based on their suitability for modeling complex, non-linear relationships in high-dimensional clinical data, their robustness to outliers and noise, and their ability to provide interpretable results. While other models like SVMs and k-NN have their merits, they present limitations in scalability, computational efficiency, and applicability to our specific dataset characteristics.

By selecting these two models, we aimed to balance performance, interpretability, and resource constraints, ensuring a thorough and meaningful analysis that could yield actionable insights in a clinical context.

  1. 2.XGBoost is indicated as the model with the best performance. Please elaborate on why this model achieved such results and how it fits the characteristics of your dataset. Discuss the specific features of XGBoost that may have contributed to its success.

XGBoost outperformed Random Forest and Neural Network models due to its unique strengths in handling complex, multi-dimensional data. Its gradient boosting mechanism allowed it to capture intricate patterns in the clinical and histopathological variables, while regularization techniques effectively controlled overfitting—a particular challenge in medical datasets.
1. Handling Complex Feature Interactions: Medical datasets often contain nonlinear relationships and intricate interactions between variables. XGBoost's gradient boosting framework builds additive models in a sequential manner, allowing it to capture these complex patterns more effectively than models like Random Forests or some Neural Networks.
2. Regularization Techniques Overfitting is a common issue in medical datasets, especially when the number of features is large compared to the number of samples. XGBoost includes L1 (Lasso) and L2 (Ridge) regularization parameters, which help in penalizing complex models and thus prevent overfitting.

We added this in the discussion part. (LINE450-452)

  1. 3.Table 1 shows significant differences in sex and BMI among recipient data. Please clarify how these differences were addressed in your analysis. Discuss whether these factors were controlled for in the models and how they might influence the outcomes.

We included these variables in both the univariate logistic regression and multivariate analyses. Based on the results, only sex significantly contributed to the model during the backward elimination process, and it has been retained in our models. In contrast, recipient BMI did not significantly improve model performance, as shown in the results table 2. Since our primary objective is to develop a predictive model, we intend to include only variables that significantly enhance model performance.

  1. 4.In Figure 5, the AUC results appear to be similar whether or not biopsy data is included. Please provide a detailed analysis of these results. Are there any statistically significant differences between the two approaches? A clearer explanation would help contextualize the findings.

Thank you for your recommendation. We have conducted DeLong’s test to evaluate the difference between the models with and without the histopathology factor. The test showed no significant difference, with a p-value of 0.246. (figure 5 )

  1. 5.The calculation of the LR+ (likelihood ratio positive) in Table 3 needs further clarification. Explain the formula used and why the values are greater than 1.

This metric assesses the probability of a positive test result in patients with the condition compared to those without it. Values greater than 1 indicate that the model has a higher likelihood of correctly identifying patients with DGF than those without it, providing strong predictive value. The LR+ values greater than 1 in our study demonstrate that the model is effective in distinguishing between patients at risk for DGF and those not at risk, supporting its potential clinical utility in decision-making.

  1. 6.Figure 7 lacks information regarding the number of tests conducted for each model. Please include this information to provide a clearer understanding of the robustness and reliability of the results.

The outer loop of the nested cross-validation involved a 10-fold split, where the data was divided into 10 parts. Each part served as a test set in turn, while the remaining 9 parts were used for training. This outer loop was repeated 10 times to ensure comprehensive model assessment across different data splits, thereby increasing result reliability.

Within each fold of the outer loop, an inner 10-fold cross-validation was conducted for hyperparameter tuning. This inner loop ensured that optimal model parameters were selected without using the test set data from the outer loop, thus preventing data leakage and preserving the integrity of the outer validation results. This process yielded a total of 100 test evaluations per model (10 outer folds repeated 10 times), which are now noted in Figure 7. This nested cross-validation approach reinforces the robustness and reliability of our reported performance metrics, providing a clear understanding of the models' stability. (LINE213-217, Figure 2)

  1. 7.More related advances (VIEW 2023, 4, 20220039; VIEW 2023, 4, 20220038) should be included for discussion.

      We cannot access above recommendation papers.

  1. 8.Image Clarity: The clarity of the images (e.g., Figure 2) needs to be improved.

We edited an image in Figure 2 and added some explanation.

Round 2

Reviewer 1 Report

Comments and Suggestions for Authors

All comments were addressed.

Author Response

"The extremely high performance metrics (e.g., XGBoost with 0.989 AUROC) suggest potential overfitting. Meanwhile, wide confidence intervals in the XGBoost results indicate model instability. I am curious authors could do additional external validation. If you cannot, kindly give us clear explanation of how the training/test split was performed. Also, expanded description of feature engineering and preprocessing steps is required"

Q: how the training/test split was performed?

A: Due to we cannot perform external validation. We used Nested cross-validation which is better than a train/test split because it provides an unbiased and reliable performance estimate by separating hyperparameter tuning from test data evaluation. The outer loop evaluates generalization performance, while the inner loop tunes hyperparameters on training subsets, avoiding overfitting and information leakage. This ensures the model's performance reflects its true predictive ability across different datasets, making it ideal for robust validation.

explain in Figure 2(page5), 2.3.3 (Page 6)

Q: expanded description of feature engineering and preprocessing steps is required

A: We rearrange the topic into EDA, Data preprocessing, feature engineering and selection which provide clearer explain about feature engineering and preprocessing. (Page 5-6)

Plus note is made of two subsections in the revised manuscript - 2.3.1 and 2.3.2 for which there is no accompanying text, which the authors also need to address.

Ans. We added text as suggested in section 2.3.1 and 2.3.2.

Note: We also deleted one of two figure7 which is redundant.